# Is COPD the Determinant Factor for Myocardial Injury and Cardiac Wall Stress in OSA Patients?

**DOI:** 10.3390/medicina59101759

**Published:** 2023-10-02

**Authors:** Athanasios Voulgaris, Kostas Archontogeorgis, Ioulianos Apessos, Nikoleta Paxinou, Evangelia Nena, Paschalis Steiropoulos

**Affiliations:** 1Department of Pneumonology, Medical School, Democritus University of Thrace, 68100 Alexandroupolis, Greece; k.archontogeorgis@yahoo.it (K.A.); nikpaxin@gmail.com (N.P.); steiropoulos@yahoo.com (P.S.); 2MSc Program in Sleep Medicine, Medical School, Democritus University of Thrace, 68100 Alexandroupolis, Greece; julianapessos@gmail.com; 3Department of Dentoalveolar Surgery, Implantology and Oral Radiology, School of Dentistry, Aristotle University of Thessaloniki, 56403 Thessaloniki, Greece; 4Laboratory of Hygiene and Environmental Protection, Medical School, Democritus University of Thrace, 68100 Alexandroupolis, Greece; enena@med.duth.gr

**Keywords:** chronic obstructive pulmonary disease, obstructive sleep apnea, overlap syndrome, cardiovascular disease, N-terminal pro-B-type natriuretic peptide, high-sensitivity cardiac troponin T

## Abstract

*Background and Objectives*: Evidence shows that COPD-OSA overlap syndrome (OS) is more frequently accompanied by cardiovascular disease (CVD) in comparison to either disease alone. The aim of the study was to explore whether patients with OS have a higher burden of subclinical myocardial injury and wall stress compared with OSA patients. *Materials and Methods*: Consecutive patients, without established CVD, underwent polysomnography and pulmonary function testing, due to suspected sleep-disordered breathing. An equal number of patients with OS (*n* = 53, with an apnea hypopnea index (AHI) > 5/h and FEV_1_/FVC < 0.7) and patients with OSA (*n* = 53, AHI > 5/h and FEV_1_/FVC > 0.7) were included in the study. The detection of asymptomatic myocardial injury and wall stress was performed via the assessment of serum high-sensitivity cardiac troponin T (hs-cTnT) and N-terminal pro-B-type natriuretic peptide (NT-proBNP), respectively. *Results*: OS patients were older (*p* < 0.001) and had worse hypoxemic parameters, namely average oxyhemoglobin saturation (SpO2) (*p* = 0.002) and time spent with SpO2 < 90% (*p* = 0.003) during sleep as well as daytime pO2 (*p* < 0.001), than patients with OSA. No difference was observed between groups in terms of Epworth Sleepiness Scale (*p* = 0.432) and AHI (*p* = 0.587). Both levels of hs-cTnT (14.2 (9.1–20.2) vs. 6.5 (5.6–8.7) pg/mL, *p* < 0.001) and NT-proBNP (93.1 (37.9–182.5) vs. 19.2 (8.3–35.4) pg/mL, *p* < 0.001) were increased in OS compared to OSA patients. Upon multivariate linear regression analysis, levels of NT-proBNP and hs-cTnT correlated with age and average SpO2 during sleep. *Conclusions*: Our study demonstrated higher levels of hs-cTnT and NT-proBNP in OS patients, indicating an increased probability of subclinical myocardial injury and wall stress, compared with OSA individuals.

## 1. Introduction

Among the most prevalent chronic respiratory diseases are chronic obstructive pulmonary disease (COPD) and obstructive sleep apnea (OSA) [1,2]. Specifically, COPD has an estimated prevalence ranging from 6.8% to 11.7% worldwide [3], while OSA affects approximately one billion individuals worldwide [4]. Differences in prevalence rates exist for both diseases according to different regions globally, as COPD is more prevalent in the Western Pacific region and OSA in China, followed by the USA, Brazil, and India [3,4]. The coexistence of OSA and COPD in the same patient is referred to as “overlap syndrome” (OS), a distinct clinical syndrome, which imposes a significant burden on afflicted patients [5]. The prevalence of OS in the general population is quite low, ranging from 1.1 to 3.6% according to the literature [1]. Nevertheless, the probability of diagnosing OS increases when OSA or COPD has been previously diagnosed [2]. Specifically, a recent systematic review on OS reported that objectively diagnosed moderate and severe OSA (apnea hypopnea index (AHI) ≥ 15/h) is present in 30 to 50% of patients with COPD [5]. Conversely, spirometry confirmed that COPD was detected in 11.9–23.2% of OSA patients [5]. 

Both COPD and OSA share common risk factors, such as an older age, male gender, obesity, tobacco smoking, and sedentary lifestyle, all of which have a negative health impact [6]. With regards to their pathophysiology, patients with either OSA or COPD exhibit various degrees of hypoxemia and/or hypercapnia during sleep according to the level of disease severity [7]. It could be assumed that the aforementioned events are often augmented when both diseases coexist in the same patient [7]. Specifically, OS leads to worse hypoxia during sleep, has more severe nocturnal oxygen desaturations (NODs), and is more often related to daytime hypoxemia and hypercapnia than COPD or OSA alone [2]. These pathophysiological phenomena along with their shared risk factors could explain the higher incidence of comorbidities, mainly cardiometabolic diseases, in OS patients [8,9].

Importantly, cardiovascular disease (CVD) more frequently coexists in both OSA and COPD patients compared with the general population [10,11,12]. Additional evidence suggests that patients with either OSA or COPD and without established CVD exhibit low-grade myocardial injury (MI), as assessed by the serum levels of high-sensitivity cardiac troponins I and T (hs-cTnI/T) [13]. Of note, circulating hs-cTn levels were found to be associated with the severity of OSA [13] and increased mortality during an acute exacerbation of COPD (AECOPD) [14]. Furthermore, N-terminal pro-B type natriuretic peptide (NT-proBNP), a cardiac biomarker which is synthesized in and released from ventricular myocytes in response to myocardial wall stress, was also studied in COPD and OSA populations. In COPD, increased levels of NT-proBNP were associated with a higher risk of future AECOPD [15]. In contrast, the evidence regarding NT-proBNP levels in OSA patients has contradictory results [16,17,18], with some studies indicating higher levels when compared with healthy controls. 

The degree of subclinical myocardial injury (hs-cTn) and increased wall stress (NT-proBNP) has not been previously examined in patients with OS. The aim of this study was to investigate whether patients with OS have higher circulating concentrations of hs-cTn and NT-proBNP compared with OSA individuals, all free from known coronary artery disease (CAD) and heart failure (HF).

## 2. Materials and Methods

### 2.1. Study Population

The study protocol was approved by the Ethics Committee of the University General Hospital Alexandroupolis (14-1-27.01.2017), and all procedures were carried out in accordance with the Helsinki Declaration of Human Rights [19]. Informed consent was provided by all patients upon enrollment in the study. 

Consecutive patients, who were diagnosed with OSA and COPD from August 2018 until January of 2020 and were free from established CVD, were included in the study. An equal number of newly diagnosed OSA patients, matched for gender and without a history of CVD, were included during the same study period. Inclusion criteria were age ≥ 40 years, current or ex-smoking status of at least 10 packs/year, and willingness to participate in the study. Exclusion criteria were as follows: previously diagnosed OSA under treatment with CPAP or alternative therapies; central sleep apnea syndrome; hypoxemic or hypercapnic respiratory failure with need for oxygen supplementation or non-invasive ventilation; pre-existing CVD (i.e., stroke, CAD, HF, and peripheral arterial disease-(PAD)); recent infection or exacerbation of COPD in the past three months; chronic kidney disease; inflammatory or autoimmune disorders; active cancer; and drug or alcohol abuse.

### 2.2. Clinical Examination

All participants were interviewed regarding their sleep habits, smoking status, alcohol intake, comorbidities, and past and current medication. Excessive daytime sleepiness (EDS) was assessed using the validated Greek version of the Epworth Sleepiness Scale (ESS) 14, a self-administered questionnaire that comprises eight questions addressing typical everyday situations [20]. Responders were asked to rate, with a score from 0 to 3, the possibility of falling asleep in each situation. EDS was defined at scores > 10 [20].

A clinical examination related to anthropometric characteristics, namely height; weight; and neck, waist, and hip circumference, was performed. Body mass index (BMI) was calculated according to the following formula: BMI = weight (in kilograms)/height (in meters)^2^. Moreover, all participants underwent a cardiac and pulmonary physical examination for signs related to heart failure, namely lung and heart auscultation and the presence of peripheral oedema. A chest X-ray and resting 12-lead electrocardiography were carried out to assess for evidence of congestive heart failure and/or previous myocardial infarction. Finally, office blood pressure measurements were performed for the diagnosis of arterial hypertension, according to guidelines [21]. 

### 2.3. Polysomnography 

Attended polysomnography (PSG) was performed according to standardized criteria [22]. The sleep recording was carried out from 22:00 to 06:00 h, and variables were recorded on a computer system (Alice^®^ 4, Philips Respironics, Murrysville, PA, USA). A standard montage including electroencephalogram, electrooculogram, electromyogram and electrocardiogram signals was used. Pulse oximetry was registered, and airflow was detected using a nasal pressure transducer and oronasal thermal flow. The thoracic cage and abdominal motion was detected using inductive plethysmography. Sleep stages, arousals, and respiratory events were scored according to standard criteria [22]. Sleep scoring was divided into rapid eye movement (REM) and non-REM (NREM) based on current American Academy of Sleep Medicine scoring rules [22]. NREM was further subclassified into sleep stages N1, N2, and N3 [22]. Apnea was defined as a ≥90% reduction in airflow for at least 10 sec. Hypopnea was defined as a ≥30% reduction in airflow for at least 10 sec in combination with an oxyhemoglobin desaturation of at least 3% or an arousal registered by the electroencephalogram. The apnea–hypopnea index (AHI) was calculated as the average number of apneas and hypopneas per hour of PSG-recorded sleep time. OSA was diagnosed as mild (AHI ≥ 5–14.9), moderate (AHI ≥ 15–29.9), or severe (AHI ≥ 30) [22].

### 2.4. Lung Function and Diagnosis of COPD

All individuals underwent an assessment of lung function via pulmonary function testing (Screenmate, Erich Jaeger GmbH & Co., Hochberg, Germany) and arterial blood gas analysis during wakefulness. A post-bronchodilator ratio of forced expiratory volume in the 1st second (FEV_1_) to forced vital capacity (FVC) of less than 70% was considered indicative of persistent airflow limitation. COPD was diagnosed in the presence of persistent airflow limitation in accordance with chronic respiratory symptoms and a compatible history of exposure to risk factors [23]. 

### 2.5. Blood Samples and Measurements

Venous blood samples were collected in a fasting state on the morning after the polysomnography. After coagulation at room temperature and centrifugation (3000 rpm for 10 min), the serum was frozen at −80 °C until processing. Biochemical parameters regarding renal and liver function, as well as glucose, C-reactive protein (CRP), and lipid profile, were measured using an automated analyzer (ADVIA 2400, Siemens, Munich, Germany). The in vitro quantitative determination of N-terminal pro-B-type natriuretic peptide and hs-cTnT was carried out by Elecsys^®^-Electro-chemiluminescence immunoassay (ECLIA), with the reference range considered at <100 pg/mL and 100–2000 pg/mL, respectively.

### 2.6. Statistical Analysis

Statistical analyses were performed using Statistical Package for the Social Sciences (SPSS), Version 21 (IBM Corp., Armonk, NY, USA). For all parameters, the normality of distribution was checked using the Shapiro–Wilk test. Quantitative data are expressed as the median (25th–75th percentiles). Categorical variables are presented as percentages. For normally distributed data, comparisons between means were determined with the Student’s t-test, and in the case of skewed distribution the non-parametric Mann–Whitney test was applied. Comparisons of percentages between groups were performed using the chi-square test. The relationship between the variables was determined using Pearson’s or Spearman’s correlation coefficient. Multivariate linear regression was used to explore the associations between NT-proBNP and hs-cTnT serum levels with anthropometric, lung function, and sleep parameters. A *p*-value of < 0.05 was considered statistically significant.

## 3. Results

In total, 53 OS patients (49 males and 4 females) patients without CVD were consecutively examined and enrolled in the selected study period. To these, 53 OSA patients (44 males and 9 females) consecutively matched for gender and a history free from CVD were added, resulting in a total of 106 (93 males and 13 females) participants. 

OS patients were older (62 (54.5–70) versus 50 (41.5–60) years, *p* < 0.001) and more obese (BMI 37 (34.5–41.5) versus 35.3 (31.1–39.2) kg/m2, *p* = 0.050) compared with OSA patients. No differences were observed between groups in the prevalence of arterial hypertension (*n* = 30 (56.6%) for OS versus *n* = 23 (43.4%) for OSA, *p* = 0.174). A comparison of the anthropometric characteristics between groups is presented in Table 1.

Patients in the OS group demonstrated worse oxygenation during sleep, as evaluated by the average oxyhemoglobin saturation during sleep (91.1 (87.9–93.2) versus 93.3 (91–94.5)%, *p* = 0.002) and percentage of time spent with oxyhemoglobin saturation < 90% (22.7 (6.9–50) versus 9 (0.8–27.9), *p* = 0.003) during sleep. A comparison of sleep parameters between groups is presented in Table 2. As expected, OS patients exhibited worse lung function than OSA patients, as assessed by FEV_1_ (65.1 (49.6–77.4) versus 100 (88.1–115)% of predicted, *p* < 0.001) and FVC (77 (61.2–90.1) versus 97 (87.8–109.5) % of predicted, *p* < 0.001). Both hs-cTnT (14.2 (9.1–20.2) versus 6.5 (5.6–8.7) pg/mL, *p* < 0.001) and NT-proBNP (93.1 (37.9–182.5) versus 19.2 (8.3–35.4) pg/mL, *p* < 0.001) serum levels were increased in OS patients compared with OSA subjects (Figure 1 and Figure 2). A comparison of laboratory parameters between groups is presented in Table 3.

In the OSA group, NT-proBNP serum levels were positively associated with age (r = 0.350, *p* = 0.001) and percentage of time with oxyhemoglobin saturation < 90% during sleep (r = 0.357, *p* = 0.009), while they were negatively associated with average (r = −0.403, *p* = 0.003) and minimum (r = −0.332, *p* = 0.015) oxyhemoglobin saturation during sleep. In the same group, hs-cTnT serum levels were positively associated with age (r = 0.340, *p* = 0.013) and BMI (r = 0.308, *p* = 0.025) and negatively associated with oxygen partial pressure (r = −0.374, *p* = 0.006); total sleep time (r = −0.287, *p* = 0.039); and average oxyhemoglobin saturation (r = −0.282, *p* = 0.041) during sleep. Among OSA patients, NT-proBNP serum levels were positively associated with hs-cTnT serum levels (r = 0.449, *p* = 0.001). There was no association between NT-proBNP or hs-cTn serum levels and laboratory parameters.

In the OS group, NT-proBNP serum levels were positively associated with age (r = 0.432, *p* = 0.001); AHI (r = 0.299, *p* = 0.029); and time with oxyhemoglobin saturation < 90% during sleep (r = 0.559, *p* < 0.001) and negatively associated with FVC (r = −0.293, *p* = 0.034) and average (r = −0.479, *p* < 0.001) and minimum (r = −0.357, *p* = 0.009) oxyhemoglobin saturation during sleep. In the same group, hs-cTnT serum levels were positively associated with age (r = 0.338, *p* = 0.013); CRP serum levels (r = 0.295, *p* = 0.032); AHI (r = 0.348, *p* = 0.011); time with oxyhemoglobin saturation < 90% during sleep (r = 0.456, *p* = 0.001); and arousal index (r = 0.465, *p* = 0.002) and negatively associated with FVC (r = −0.365, *p* = 0.007); FEV_1_ (r = −0.289, *p* = 0.036); oxygen partial pressure (r = −0.313, *p* = 0.023); and average (r = −0580, *p* < 0.001) and minimum (r = −0.379, *p* = 0.005) oxyhemoglobin saturation during sleep. Among OS patients, NT-proBNP serum levels were positively associated with hs-cTnT serum levels (r = 0.536, *p* < 0.001).

Multivariate linear regression analysis was performed in order to demonstrate the association of NT-proBNP and hs-cTnT serum levels with anthropometric parameters (age; BMI; and neck, waist, and hip circumference); pulmonary function (FEV_1_, FVC, and PaO2); and sleep parameters (total sleep time, arousal index, AHI, average and minimum oxyhemoglobin saturation during sleep, and percentage of time with oxyhemoglobin saturation < 90% during sleep). The results revealed that NT-proBNP serum levels were positively associated with age (β = 0.194, *p* = 0.044) and negatively associated with average oxyhemoglobin saturation (β = −0.910, *p* < 0.001) during sleep. Moreover, hs-cTnT serum levels were positively associated with age (β = 0.194, *p* = 0.039) and negatively associated with average oxyhemoglobin saturation (β = −0.745, *p* < 0.001) during sleep.

## 4. Discussion

The present study reported higher levels of hs-cTnT and NT-proBNP in OS patients compared with OSA patients without known CVD. Moreover, an independent association between those markers and age and nocturnal hypoxia, as assessed by the average oxyhemoglobin saturation during sleep, was observed. To the best of our knowledge, this was the first study to examine the degree of subclinical myocardial injury and wall stress in OS as compared with OSA. 

### 4.1. Cardiovascular Comorbidity in Overlap Syndrome

Accumulating evidence suggests a close relationship between CVD and OS [2,6]. The latest data indicate a higher prevalence of cardiovascular (CV) comorbidities in OS compared with OSA or COPD alone [8,12]. A study including 163 OS patients and an equal number of OSA patients matched for confounders found that OS individuals had more comorbidities (*p* = 0.033) and a higher prevalence of CVD (*p* = 0.016) than patients with OSA [8]. Similarly, a retrospective study by Tang et al. demonstrated that patients with OS were afflicted more often by pulmonary hypertension (PH) and HF and had worse all-cause mortality than patients with COPD or OSA (all *p* < 0.05) [9]. Finally, a recent systematic review [12], including patients with OS (n = 4613), OSA (*n* = 16,046), and COPD (*n* = 1679), showed that OS was associated with a higher risk of hypertension than COPD and an increased risk of CAD, HF, and stroke compared to OSA.

### 4.2. NT-proBNP and hs-cTn as Markers of Myocardial Wall Stress and Injury 

There is an increasing need to find biomarkers in clinical practice for CV assessment. NT-proBNP and hs-cTn are readily available biomarkers in this domain, allowing the reliable assessment of the cardiac status in a fast and noninvasive way. BNP is a cardiac neurohormone released primarily from the ventricles in response to volume expansion, myocyte stretching, and possibly increased wall stress [24]. The prohormone of BNP is the molecule pro-BNP, which is cleaved biologically into an active 32-amino-acid BNP and a 76-amino-acid N-terminal pro-BNP (NT-proBNP) [25]. NT-proBNP is increased in several cardiac diseases, like HF, CAD, and myocardial hypertrophy, reflecting left-ventricular (LV) dysfunction [26,27]. In addition, cardiac troponin is released from cardiac myocytes following myocardial damage and is therefore an established marker of acute coronary syndrome [28]. Importantly, hs-cTnT/I assays are able to detect very low circulating troponin concentrations and could be used for risk stratification in patients at risk for CV events and primary preventive treatment [29]. 

### 4.3. Assessment of NT-proBNP and hs-cTn in OSA and COPD Populations 

In our study, the serum levels of NT-proBNP and hs-cTnT were elevated in a group of patients with OS and without clinically evident CVD compared with OSA individuals. Previous studies have assessed these cardiac markers in OSA and COPD populations. A prospective study assessing hs-cTn in 1599 patients with COPD over a period of 18 months showed that those with the highest concentrations of hs-cTnI were at greater risk of CV events and death after adjustment for confounding factors [30]. Of note, the majority of patients included in this study exhibited levels of hs-cTnI above the limit of detection. Similarly, the multicenter COSYCONET cohort study, including patients with stable COPD, demonstrated that hs-cTnI was a significant predictor of all-cause mortality after adjustment for CV risk factors [31]. Finally, a study including 99 patients hospitalized due to AECOPD found that those with hs-cTnT above the normal range (≥14.0 ng/L) had an increased risk of mortality compared to those with normal-range hs-cTnT levels [14].

With regards to OSA, an analysis of 1645 participants from multicenter studies, of whom 65% had detectable levels of hs-cTnT, showed that levels of hs-cTnT were associated with an increasing severity of OSA, as assessed by the respiratory disturbance index (RDI), as well as the risk of death and incident HF in OSA patients, irrespective of the RDI [32]. Another study from the Akershus Sleep Apnea Project found that 62% of patients had a hs-TnI concentration above the limit of detection (1.2 ng/L), and that higher hs-cTnI values were related to higher AHI values and worse nocturnal hypoxic indices [13]. Overall, existing data indicate detectable concentrations of hs-cTn in OSA and COPD populations and reveal important clinical associations for hs-cTn in these patients. 

Concerning NT-proBNP, an analysis of SPIROMICS, assessing 1051 participants with stable COPD and a mean NT-proBNP level of 608.9 pg/mL, showed that baseline NT-proBNP was associated with an increased risk of incident exacerbations, irrespective of coexistent CVD [15]. A recent meta-analysis summarized the evidence on NT-proBNP levels across distinct patient groups with COPD [33]. It showed that stable COPD patients had increased levels of NT-proBNP compared to controls, and that NT-proBNP levels were higher in acute exacerbations compared to patients with stable disease. COPD patients with concurrent HF or pulmonary hypertension had worse levels than individuals with COPD alone [33]. 

Studies on OSA populations showed opposing results regarding NT-proBNP values. Tasci et al. [16] found no differences in levels of pro-BNP between OSA and healthy subjects, whereas CPAP reduced the levels of pro-BNP in both normotensive and hypertensive OSA patients. Another study evaluating 60 patients with suspected OSA found no correlations between NT-proBNP and AHI, hypoxic parameters during sleep, and daytime sleepiness [17]. Nevertheless, treatment with CPAP reduced NT-proBNP levels at 3 months but only in those patients with pathologically elevated levels of NT-proBNP at baseline [17]. On the contrary, one study evaluated 1292 patients with CAD and suspected OSA [34]. Included patients had a mean NT-proBNP value of 826.57 μg/L, and those with higher levels of NT-proBNP had a greater likelihood of comorbid OSA [34]. Considering these findings, our results indicate that the additive value of COPD in OSA patients could explain why OS patients showed higher levels of NT-proBNP compared with the OSA patient group.

### 4.4. Associations between hs-cTnT and NT-proBNP and Parameters Related to OSA and COPD

We found several associations between serum levels of hs-cTnT and NT-proBNP with COPD and OSA related-factors. Specifically, NT-proBNP correlated with nocturnal hypoxic indices in both groups, and additionally with age, AHI and FVC in OS. Moreover, at multivariate linear regression analysis, NT-proBNP levels were independently associated with age and average oxyhemoglobin saturation during sleep. Our findings highlight age, and severity of nocturnal hypoxia related to OSA as significant risk factors for myocardial wall stress. Evidence acknowledges hypoxemia as an important stimulus for NT-proBNP release [35], while greater age is related to higher NT-proBNP levels [36] and increased risk for heart failure [37,38]. In the present study OS patients were older and had worse average oxyhemoglobin saturation during sleep than OSA subjects. In summary, our results could explain the differences in NT-proBNP values between the two groups. 

Both groups also showed associations between hs-cTnT and age, daytime and nocturnal hypoxemia. In the group of patients with OS hs-cTnT levels were related additionally with AHI and FEV_1_ and FVC, among others. In the whole group and after adjustment for confounders, age and average oxyhemoglobin saturation during sleep were again revealed as correlates of hs-cTnT levels. These data also highlight significant interactions between circulating levels of hs-cTn and age and nocturnal hypoxia. Factors associated with hs-cTn release could be hypoxemia, hypecapnia and comorbid pulmonary hypertension in advanced stages of COPD, which is characterized by reduced FEV1 [39]. Of note, all aforementioned factors are more heightened during an AECOPD [40]. Moreover, OSA is characterized by repetitive episodes of intermittent hypoxemia and surges of blood pressure during sleep, which impose a risk of myocardial injury and elevated levels of hs-cTn, as well [2]. Hypoxemia during sleep is present in OSA and COPD, and is even more profound in OS [7]. In fact, OS patients exhibit greater nocturnal oxygen desaturation, as evidenced by the time spent with oxygen saturation < 90% and the average oxyhemoglobin saturation during sleep, than OSA individuals [1]. Finally, as noted before age constitutes one of the strongest risk factors for CAD [41] and it was also found to independently correlate with hs-cTn levels. The latter was present in our study and could clarify the correlation found between age, average oxyhemoglobin saturation during sleep and hs-cTnT. 

### 4.5. Limitations 

Our study is subject to several limitations. Firstly, included patients with OS were older compared with OSA individuals, and this factor has influenced the study results. Specifically, age is an independent risk factor for CVD and likewise our study showed that age correlated with both cardiac markers. Secondly, the number of included female patients was small and thus the study results should be interpreted with caution. Nevertheless, both groups showed no differences in the male to female ratio. Another potential weakness of this study was that concentrations of hs-cTn were assessed via hs-TnT assay. There is evidence to suggest that the hs-TnI assay may be a superior tool to detect the troponin signal than the hs-TnT assay [42]. Nevertheless, both assays offer a reliable assessment of CVD death and heart failure [42]. In addition, the OS group comprised expectedly of a greater proportion of patients with smoking history and had higher BMI values than the OSA group. It is well known the effects of smoking [43] and obesity [44,45] increase the risk of CVD independently of the presence of COPD and OSA, respectively. However, after controlling for confounding factors, neither smoking history nor BMI independently correlated with hs-TnT and NT-proBNP serum levels in the present study. Finally, a major limitation of our study is the lack of echocardiography measurements, which could have provided precise assessments of left ventricular size and function and might have precisely explained the differences in levels of NT-proBNP between the two groups. Future studies implementing echocardiography together with measurement of these cardiac markers would provide further insight on the risk of CVD development in OS. 

## 5. Conclusions

In conclusion, our results suggest that both levels of hs-cTn and NT-proBNP are increased in a group of OS patients compared with OSA individuals and without CVD, indicating a higher probability of subclinical myocardial injury and wall stress. Further research is necessary in order to elucidate the impact of overlap syndrome on the development of subclinical CVD and to implement preventive strategies towards this approach. 

## Figures and Tables

**Figure 1 medicina-59-01759-f001:**
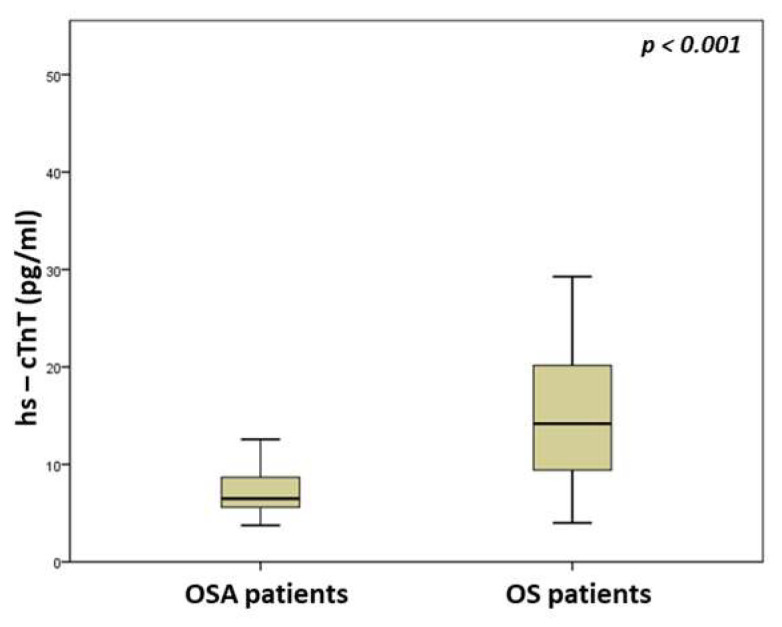
Comparison of high-sensitivity troponin T serum levels between OSA and OS patients.

**Figure 2 medicina-59-01759-f002:**
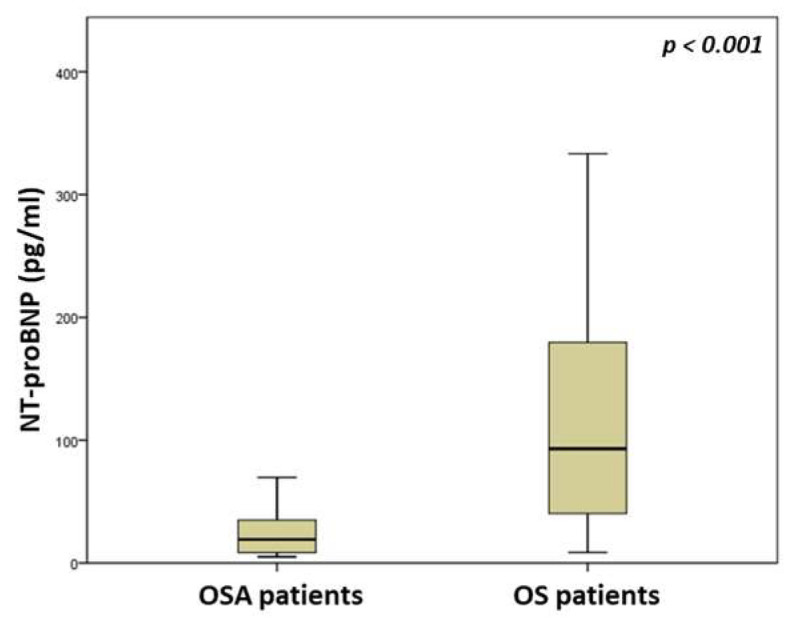
Comparison of N-terminal pro-B type natriuretic peptide serum levels between OSA and OS patients.

**Table 1 medicina-59-01759-t001:** Comparison of anthropometric characteristics between obstructive sleep apnea (OSA) and overlap syndrome (OS) patients.

	OSA Patients (*n* = 53)	OS Patients (*n* = 53)	*p*
Gender (males/females)	44/9	49/4	0.139
Age (years)	50 (41.5–60)	62 (54.5–70)	<0.001
Neck circumference (cm)	44 (42–46)	46.5 (44–50)	<0.001
Waist circumference (cm)	118 (110–126)	132 (120–136)	<0.001
Hip circumference (cm)	115 (108–123)	119 (111–128)	0.156
WHR	0.88 (0.82–0.97)	0.90 (0.87–1.01)	0.158
BMI (kg/m^2^)	35.3 (31.1–39.2)	37 (34.5–41.5)	0.050
Tobacco smoking	62.3%	86.8%	0.004
Alcohol consumption	64.2%	73.6%	0.294

Abbreviations: BMI—body mass index, WHR—waist to hip ratio.

**Table 2 medicina-59-01759-t002:** Comparison of sleep parameters between obstructive sleep apnea (OSA) and overlap syndrome (OS) patients.

	OSA Patients (*n* = 53)	OS Patients (*n* = 53)	*p*
TST (min)	337.3 (301.3–362.1)	317.8 (281.5–351.5)	0.132
N1 (%)	5.2 (3.5–9.5)	9.9 (4.5–18.2)	0.011
N2 (%)	78.8 (72.3–89)	71.2 (63.1–85.9)	0.004
N3 (%)	4.3 (0–10.7)	7.4 (0–15.5)	0.119
REM (%)	4.9 (1.7–9.5)	4 (0.8–11.5)	0.824
AHI (events/hour)	36.5 (19.1–66.8)	35.5 (18–53.5)	0.587
Aver SpO2 (%)	93.3 (91–94.5)	91.1 (87.9–93.2)	0.002
Min SpO2 (%)	72 (65–84)	75 (65.5–80)	0.695
T < 90% (%)	9 (0.8–27.9)	22.7 (6.9–50)	0.003
Arousal index	1 (0–4)	5 (1–7.8)	0.001
Sleep efficiency (%)	89.1 (79–93.3)	84 (74.6–89.7)	0.027
ESS score	10 (6.5–13)	11(6–16)	0.432

Abbreviations: AHI—apnea hypopnoea index, Aver SpO2—average oxyhemoglobin saturation, ESS—Epworth sleepiness scale, Min SpO2—minimum oxyhemoglobin saturation, N1—sleep stage 1, N2—sleep stage 2, N3—sleep stage 3, REM—rapid eye movement, TST—total sleep time, T < 90%—time with oxyhemoglobin saturation < 90%.

**Table 3 medicina-59-01759-t003:** Comparison of laboratory parameters and lung function between obstructive sleep apnea (OSA) and overlap syndrome (OS) patients.

	OSA Patients (*n* = 53)	OS Patients (*n* = 53)	*p*
hs-cTnT (pg/mL)	6.5 (5.6–8.7)	14.2 (9.1–20.2)	<0.001
NT-proBNP (pg/mL)	19.2 (8.3–35.4)	93.1 (37.9–182.5)	<0.001
CRP (mg/dl)	0.49 (0.15–0.81)	0.48 (0.16–0.81)	0.246
Glucose (mg/dL)	111 (97–141)	112 (93.5–133.8)	0.703
Creatinine (mg/dL)	0.9 (0.8–1)	0.9 (0.85–1.1)	0.206
SGOT (U/L)	20 (18–25)	20 (17–23.8)	0.780
SGPT (U/L)	26 (19–34.5)	22 (18–27)	0.131
Cholesterol (mg/dL)	201 (185.5–226)	189 (162–224)	0.199
Triglycerides (mg/dL)	181 (133.5–206)	156 (118–216.5)	0.200
LDL-C (mg/dL)	119.5 (99.3–144.8)	104.6 (80.4–141.6)	0.123
HDL-C (mg/dL)	46 (40–54)	46 (40.5–51)	0.914
FEV_1_ (% predicted)	100 (88.1–115)	65.1 (49.6–77.4)	<0.001
FVC (% predicted)	97 (87.8–109.5)	77 (61.2–90.1)	<0.001
FEV_1_/FVC (%)	85.7 (81.5–89)	68.9 (63.1–69.7)	<0.001
pH	7.43 (7.42–7.44)	7.42 (7.41–7.44)	0.072
pO2 (mmHg)	80 (72.5–87)	68 (61–78)	<0.001
pCO_2_ (mmHg)	42 (40–43)	46 (42–51)	<0.001

Abbreviations: FEV_1_—forced expiratory volume in 1st sec, FVC—forced vital capacity, HDL-C—high-density lipoprotein-cholesterol, hs-cTnT—high-sensitivity troponin T, LDL-C—low-density lipoprotein-cholesterol, pCO_2_—carbon dioxide partial pressure, pO2—oxygen partial pressure, NT-proBNP—N-terminal pro-B type natriuretic peptide, SGOT—serum glutamic oxaloacetic transaminase, SGPT—serum glutamic pyruvic transaminase.

## Data Availability

The data relevant to this study are presented in this article.

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
