# Peer review of "Is COPD the Determinant Factor for Myocardial Injury and Cardiac Wall Stress in OSA Patients?"

_medicina, 2023, doi:10.3390/medicina59101759_

Round 1

Reviewer 1 Report

The authors have shown Overlap syndrome as a link between COPD and OSA and its implications in myocardial injury. The study is exciting and have promising outcomes. However, there are few recommendations:

1.    Is there any data about the problem associated with different regions or ethnicities globally? Mention in the introduction section. Wherever necessary, please provide latest information/data for the COPD, OSA and CVDs.

2.    The patients were considered for the study between Aug 2018 and January 2020. Did the authors think that there would be further variations in post-covid era?

3.    The criteria to consider the male to female ratio is unclear. The sample size could be better.

4.    Did the authors check the inflammatory pathways as key biomarkers of circulating inflammatory markers that could have correlated accurately differentiated between the groups. The serum content may be used to study the aforementioned.

5.    Alcohol consumption increases the risk of sleep apnoea. Why does it not be included as a parameter in Table 1? Any note on Hypertension ?

6.    It would be important to explain briefly about N1-N3 sleep stages to make it more understandable in context to the current problem.

7.    Did the authors do the baseline levels of NT-proBNP and hs-cTn? How was the comparison done? Why were no healthy subjects recruited for the study?

8.    Fig.1 and Fig.2 must be improved with the figure legends to explain the parameters. The standard error is so high. It would be important to include bigger sample size.

Author Response

The authors have shown Overlap syndrome as a link between COPD and OSA and its implications in myocardial injury. The study is exciting and have promising outcomes. However, there are few recommendations:

Reply: We would like to thank you for your insightful comments and suggestions. We hope that the revision satisfactorily addresses your comments.

  1. Is there any data about the problem associated with different regions or ethnicities globally? Mention in the introduction section. Wherever necessary, please provide latest information/data for the COPD, OSA and CVDs.

Reply: Thank you for this comment. Indeed, there are differences in prevalence rates in OSA and COPD according to different regions globally [1,2]. We have now added the most recent studies regarding the prevalence of COPD and OSA in the introduction section.

  1. The patients were considered for the study between Aug 2018 and January 2020. Did the authors think that there would be further variations in post-covid era?

Reply: Thank for your comment. Our study was designed to be completed before the start of COVID-19 pandemic. We think post-pandemic data is an important field for future research, as patients with COPD or OSA are more susceptible to infection with SARS-CoV-2 and experience worse outcomes related to COVID-19 [3,4]. Thus, cardiovascular comorbidity could be heightened following COVID-19.

  1. The criteria to consider the male to female ratio is unclear. The sample size could be better.

Reply: Thank for your comment. We had initially aimed to include an equal number of consecutive patients with overlap syndrome and OSA without known CVD for the duration of the study period. Moreover, another aim was to enroll approximately the same number of males and females, so as to eliminate differences between groups in terms of gender. Nevertheless, the overall prevalence of overlap syndrome is generally low and there is also an overrepresentation of male patients in both OSA and COPD, making it difficult to enroll female patients with overlap syndrome- especially without CVD. We are aware of this limitation, and we have addressed this issue in the limitation section of diacussion. Future large multicenter studies could overcome this enrollment barrier and assess properly the risk of CVD in female patients with overlap syndrome.

  1. Did the authors check the inflammatory pathways as key biomarkers of circulating inflammatory markers that could have correlated accurately differentiated between the groups? The serum content may be used to study the aforementioned.

Reply: Thank you for this comment. This is a very important aspect to consider in order to better understand the pathophysiological mechanisms linking OS, OSA, COPD and CVD. Unfortunately, we had aimed to assess only the serum levels of NT-proBNP and hs-cTnT as well as C-reactive protein, apart from the routine laboratory evaluation.

  1. Alcohol consumption increases the risk of sleep apnea. Why does it not be included as a parameter in Table 1? Any note on Hypertension ?

Reply: Thank you for this constructive comment. Unfortunately, we have now included in our results the prevalence of alcohol consumption and arterial hypertension for both groups.   

  1. It would be important to explain briefly about N1-N3 sleep stages to make it more understandable in context to the current problem.

Reply: Thank you for this constructive comment. We have now included in the Methods section (polysomnography) an explanation regarding the classification of sleep stages.  

  1. Did the authors do the baseline levels of NT-proBNP and hs-cTn? How was the comparison done? Why were no healthy subjects recruited for the study?

Reply: Thank you for this comment. The aim of this study had been to assess the baseline levels of NT-proBNP and hs-cTnT in newly diagnosed patients with OS and OSA and before initiation of any treatments (as seen in Table 1 and in lines 200-202). Moreover, earlier studies have shown that either OSA or COPD are associated with higher serum levels of NT-proBNP and hs-cTnT compared with healthy individuals. Considering this, we aimed to directly compare these patient groups without including healthy controls, as this aspect was already shown in previous studies. Comparisons were made with the students t test.

  1. 1 and Fig.2 must be improved with the figure legends to explain the parameters. The standard error is so high. It would be important to include bigger sample size.

Reply: Thank you for this comment. We have updated the figure legends and made them clearer. Indeed, the standard error is high, and this is attributable to the small sample size included in our study. We are aware of this limitation, and we have mentioned this issue in the limitations part of Discussion. Our explanation is that prevalence of overlap syndrome is lower than each disease alone [5] and not so frequent to enroll patients with overlap syndrome and without CVD in a single center. In fact, CVD is highly prevalent in patients with OS at the time of their initial evaluation [6]. We hope that our study will guide future research in order to assess the burden of myocardial disease in patients with OS even when CVD is not clinically evident.  

References

[1]          Adeloye D, Song P, Zhu Y, Campbell H, Sheikh A, Rudan I, et al. Global, regional, and national prevalence of, and risk factors for, chronic obstructive pulmonary disease (COPD) in 2019: a systematic review and modelling analysis. Lancet Respir Med 2022;10:447–58. https://doi.org/10.1016/S2213-2600(21)00511-7.

[2]          Benjafield AV, Ayas NT, Eastwood PR, Heinzer R, Ip MSM, Morrell MJ, et al. Estimation of the global prevalence and burden of obstructive sleep apnoea: a literature-based analysis. Lancet Respir Med 2019;7:687–98. https://doi.org/10.1016/S2213-2600(19)30198-5.

[3]          Aveyard P, Gao M, Lindson N, Hartmann-Boyce J, Watkinson P, Young D, et al. Association between pre-existing respiratory disease and its treatment, and severe COVID-19: a population cohort study. Lancet Respir Med 2021;9:909–23. https://doi.org/10.1016/S2213-2600(21)00095-3.

[4]          L Mandel H, Colleen G, Abedian S, Ammar N, Charles Bailey L, Bennett TD, et al. Risk of post-acute sequelae of SARS-CoV-2 infection associated with pre-coronavirus disease obstructive sleep apnea diagnoses: an electronic health record-based analysis from the RECOVER initiative. Sleep 2023;46:zsad126. https://doi.org/10.1093/sleep/zsad126.

[5]          Shawon MSR, Perret JL, Senaratna CV, Lodge C, Hamilton GS, Dharmage SC. Current evidence on prevalence and clinical outcomes of co-morbid obstructive sleep apnea and chronic obstructive pulmonary disease: A systematic review. Sleep Med Rev 2017;32:58–68. https://doi.org/10.1016/j.smrv.2016.02.007.

[6]          Shah AJ, Quek E, Alqahtani JS, Hurst JR, Mandal S. Cardiovascular outcomes in patients with COPD-OSA overlap syndrome: A systematic review and meta-analysis. Sleep Med Rev 2022;63:101627. https://doi.org/10.1016/j.smrv.2022.101627.

Reviewer 2 Report

The manuscript of Dr Voulgaris and colleagues is very interesting, because, the authors clearly demonstrate that in Overlap patients (COPD+OSA) there is a higher risk to develop a myocardial injury, than in OSA subjects alone.

However, in my opinion there are some limitations that the authors should underline in the manuscript.

First: Increased levels of NT-proBNP, and of hs-c TnT are expected to be higher in Overlap patients, than in OSA, probably due to the effect of smoking on the myocardial tissue.

Second: It has been recently demonstrated, by a couple of papers, that the cardiovascular risk is definitely increased in patients with higher BMI, indepedently by the severity of OSA, demonstrating that the systemic inflammation due to obesity plays a primary role in determining myocardial diseases ( DOI: 10.1016/J.Jstrokecerebrovascdis.2023.107221 and DOI: 10.1111/IJCP.14952 ) .

Minor: English should be revised by a native speaker.

English should be revised by a native speaker.

Author Response

Comments and Suggestions for Authors

The manuscript of Dr Voulgaris and colleagues is very interesting, because the authors clearly demonstrate that in Overlap patients (COPD+OSA) there is a higher risk to develop a myocardial injury, than in OSA subjects alone.

Reply: We would like to thank you for your insightful comments and suggestions. We hope that our revised paper addresses your comments satisfactorily.

However, in my opinion there are some limitations that the authors should underline in the manuscript.

First: Increased levels of NT-proBNP, and of hs-c TnT are expected to be higher in Overlap patients, than in OSA, probably due to the effect of smoking on the myocardial tissue.

Reply: Thank you for this comment. We agree with your comment, and we have addressed this in the limitations section of the revised paper.

Second: It has been recently demonstrated, by a couple of papers, that the cardiovascular risk is definitely increased in patients with higher BMI, independently by the severity of OSA, demonstrating that the systemic inflammation due to obesity plays a primary role in determining myocardial diseases ( DOI: 10.1016/J.Jstrokecerebrovascdis.2023.107221 and DOI: 10.1111/IJCP.14952 ) .

Reply: Thank you for this comment. We do agree with your comment, and we have addressed this issue in the limitation section of the revised paper. Moreover, we have cited in the revision the aforementioned references.  However, in our sample we did not find any associations between BMI and the serum levels of hs-cTnT and NT-proBNP after controlling for confounding factors. 

Minor: English should be revised by a native speaker.

Reply: Thank you for this suggestion. The revised paper is now proof edited by a native speaker.